# Multilingual Fact Linking

**Keshav Kolluru** [1] *                                      KESHAV.KOLLURU@GMAIL.COM

**Martin Rezk** [2]                                                      MREZK@GOOGLE.COM

**Pat Verga** [2]                                                            PAT@GOOGLE.COM

**William Cohen** [2]                                                  WCOHEN@GOOGLE.COM

**Partha Talukdar** [2]                                               PARTHA@GOOGLE.COM

[1] *Indian Institute of Technology, Delhi*
[2] *Google*

## Abstract

Knowledge-intensive NLP tasks can benefit from linking natural language text with facts from a Knowledge Graph (KG). Although facts themselves are language-agnostic, the *fact labels* (i.e., language-specific representation of the fact) in the KG are often present only in a few languages. This makes it challenging to link KG facts to sentences in languages other than the limited set of languages. To address this problem, we introduce the task of Multilingual Fact Linking (MFL) where the goal is to link fact expressed in a sentence to corresponding fact in the KG, even when the fact label in the KG is not available in the language of the sentence. To facilitate research in this area, we present a new evaluation dataset, INDICLINK. This dataset contains 11,293 linked WikiData facts and 6,429 sentences spanning English and six Indian languages. We propose a Retrieval+Generation model, REFCOG, that can scale to millions of KG facts by combining Dual Encoder based retrieval with a Seq2Seq based generation model which is constrained to output only valid KG facts. REFCOG outperforms standard Retrieval+Re-ranking models by 10.7 pts in Precision@1. In spite of this gain, the model achieves an overall score of 52.1, showing ample scope for improvement in the task. REFCOG code and INDICLINK data are available at github.com/SaiKeshav/mfl

## 1. Introduction

Knowledge Graphs (KG) are a large and useful source of information about the world and are composed of curated facts regarding entities. Each fact asserts a relation that exists between two entities, which can be expressed as *(subject; relation; object)*. KGs have proven to be helpful for NLP tasks like Question Answering [Saxena et al., 2020] and are used in a variety of industry applications [Dong, 2017, Fensel et al., 2020]. KG facts represent knowledge explicitly in a structured format that allows for greater interpretability. Linking KG facts with text has several benefits. Fact linking can potentially be used by systems to verify claims and detect misinformation. Moreover, KG facts allow for rich search experiences by supporting knowledge panels which require triggering the correct panels by connecting search queries to KG facts. Recent works [Verga et al., 2021] have shown that augmenting pretrained transformers with KG facts allow for modular management of knowledge that can be tuned according to the end application, such as adding new facts or removing stale ones. Improved fact-linked text can benefit such models even further.

---

*. Work done at Google.

Knowledge graphs such as Wikidata[1] often contain entity and property labels in multiple languages (although with severe skew [Kaffee et al., 2017]). For example, *Argovia* and *Argau* are the English and Spanish *entity labels* for entity Q11972 in Wikidata. Similarly, *country* and *país* are the English and Spanish *property labels* for property P17. Combining these, we define the notion of *fact labels* which are language-specific representations of an otherwise language-agnostic fact. For example, *(Aargau; country; Switzerland)* and *(Argovia; país; Suiza)* are the English and Spanish fact labels for the fact (Q11972; P17; Q39) in Wikidata.

While linking KG facts to text has received some attention in literature [Elsahar et al., 2018], most of that work has been restricted to English fact labels and text. There is a growing need to connect fact and their mentions in text, especially when languages of the fact label and text are different. For example, we would like to link a fact with an English fact label with a text in Hindi or Telugu. We refer to this problem as **Multilingual Fact Linking (MFL)**. We define the task as predicting all the KG facts that are implied by a sentence even when the the sentence and KG fact labels are present in different languages. To the best of our knowledge there has been no prior work on this important problem.

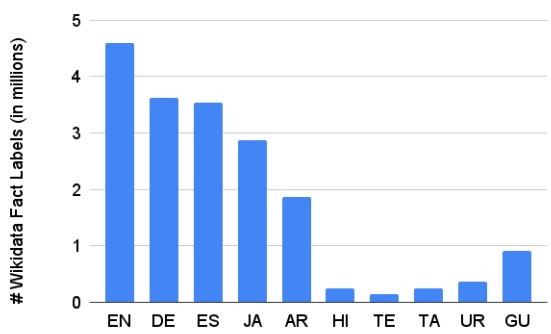

Figure 1: Distribution of languages of fact labels (in millions) on a subset of Wikidata. Compared to English and a few other languages, fact labels in Indian languages (last five: HI, TE, TA, UR, GU) are extremely sparsely represented. We focus on Multilingual Fact Linking, especially in the context of Indian languages, in this paper.

This challenging problem is made further difficult due to language skew among the fact labels in a KG. In Figure 1, we present the language distribution of fact labels (across ten languages) obtained from a subset of Wikidata. We find that the fact labels are heavily-skewed towards higher-resources languages (first five), compared to the remaining five Indian languages, viz., Hindi (HI), Telugu (TE), Tamil (TA), Urdu (UR), and Gujarati (GU). These languages are spoken by hundreds of millions of speakers, although they are severely understudied in the NLP community. We focus on the problem of MFL, with particular emphasis on Indian languages, in this paper.

To facilitate research on the problem of multilingual fact linking, we curate a new evaluation dataset, **INDICLINK**, containing parallel sentences in English and six Indian languages tagged with the corresponding Wikidata facts expressed in them. The English sentences and facts are taken from the relation extraction dataset, WebRED [Ormándi et al., 2021]. The test examples are manually translated into different languages and automatic translation of sentences are used for training. We use KG facts from Wikidata and explore different strategies to use their fact labels in English and other languages, wherever available.

Since multilingual fact linking involves selecting a subset of facts from the entire set of KG facts, the label space may be in the scale of millions. To handle this, we make use of Dual Encoder-Cross Encoder paradigm, where a dual encoder [Reimers and Gurevych, 2019b] is used to quickly retrieve

---

1. https://www.wikidata.org/

| Task | Input/Output |
|---|---|
| Entity Linking [Botha et al., 2020] | Table Jura stretches across the Swiss cantons of {*Basel-Landschaft*} and Aargau. **Output**: Q12146: *Landschaft* |
| Relation Classification [Ormándi et al., 2021] | Table Jura stretches across the OBJ{*Swiss*} cantons of SUBJ{*Basel-Landschaft*} ... **Output**: P17: *country* |
| Fact Extraction [Elsahar et al., 2018] | {*Table Jura*} stretches across the {*Swiss*} cantons of Basel-Landschaft and Aargau. **Output**: (Q356545: *Table Jura*; P17: *country*; Q39: *Switzerland*) |
| Multilingual Fact Linking (This Paper) | टेबल जुरा बासेल-लैंडशाफ्ट और आर्गो के स्विस कैन्टन में फैला हुआ है। (*tebal jura baasel-laindshaaft aur aaragau ke svis kaintan mein phaila hua hai*) **Output**: $F_{23}$ = (Q12146: *Landschaft*; P17: *country*; Q39: *Switzerland*) $F_{52}$ = (Q11972: *Aargau*; P17: *country*; Q39: *Switzerland*) |

Table 1: KG linking task examples. Multilingual Fact linking involves discovering the subset of KG facts expressed in a sentence, even when fact labels are available in a different language, requiring cross-lingual inference (Hindi-English in above example). Fact Linking systems only output facts already present in the KG. Fact Extraction aims to discover new facts not present in the KG, such as (*Table Jura*, *country*; *Switzerland*) in the above example. *Q* and *P* represent the entity and property identifiers in Wikidata. The fact identifiers (e.g., $F_{23}$) are assigned and are not part of Wikidata.

the potential top–$k$ facts. These top–$k$ facts are typically re-ranked using classification based cross encoder architectures. However, due to the pipelined nature of the system, performance of the re-ranking model is limited by that of the retrieval model. We propose a novel model, **Re**trieval based **F**act-**Co**nstrained **G**eneration (**ReFCoG**), which replaces *re-ranking* cross encoders with a Seq2Seq based cross encoder that is constrained to *generate* facts from the KG, even those which are absent in the retrieved set of facts. We show that such generative models outperform classification based re-ranking, achieving 10.7 pts improvement in precision.

In summary, the main contributions of this work are as follows:

1. We introduce the task of Multilingual Fact Linking to link KG facts with their mentions in text, especially when there is a mismatch between the languages of the fact label and text.

2. We present INDICLINK, an evaluation dataset for MFL in English and six Indian languages, a widely used but less studied set of languages in the NLP community. To the best of our knowledge, this is the first dataset of its kind for these languages.

3. We propose REFCOG, a novel Retrieval+Generation model for the task of MFL. The proposed method significantly outperforms standard Retrieval+Re-Ranking models.

## 2. Related Work

External sources of knowledge are helpful for NLP tasks such as question answering and fact verification. KILT [Petroni et al., 2020] uses a retrieval+seq2seq model with Wikipedia documents as the knowledge source to solve many knowledge-intensive tasks. FAE [Verga et al., 2021] uses WikiData KG to inject knowledge into pretrained language models in a modular fashion. Colon-Hernandez et al. [2021] surveys various methods of combining structured knowledge such as KGs and pretrained language models. However, they don't handle multilingual examples in both the text or KG. Linking text to KGs has been traditionally explored in various settings, such as entity linking, relation classification and extraction, fact extraction and fact linking. We list various KG-related tasks and their corresponding inputs and outputs in Table 1.

**Entity Linking**: Entity linking involves finding the KG entity referred by a mention marked in the input sentence. Multilingual pretrained models have advanced Entity Linking across languages. MEL [Botha et al., 2020] uses a Dual Encoder, Cross Encoder pipeline trained on hard negatives to achieve strong performance on 104 languages. mGENRE [Cao et al., 2021] is an entity linking model that autoregressively generates the linked entity name using a Seq2Seq model. Moreover, it augments the decoder with a prefix trie in order to generate only valid entity names. We find that combining both of these models by using a Dual Encoder, Constrained Generation pipeline leads to strong multilingual fact linking performance.

**Relation Classification**: The task involves identifying the relation between a pair of entity mentions. It is also referred to as relation extraction. The recently released WebRED [Ormándi et al., 2021] provides a dataset of English sentences annotated with the corresponding Wikidata relation/predicate and the linked entities. We create INDICLINK using these examples. Other multilingual relation classification datasets such as RELX [Koksal and Ozgur, 2020] are unusable for fact alignment as they don't provide the linked entities.

**Fact Extraction**: The task involves joint entity and relation extraction [Zhong and Chen, 2021, Sui et al., 2020] that focus on discovering new facts that are not present in the KG. Whereas, fact linking deals with connecting existing KG facts with text. Therefore, fact linking models make use of KG facts (which maybe in the scale of millions) whereas fact extraction systems do not.

**Fact Linking**: Existing fact linking/alignment systems such as T-REx [Elsahar et al., 2018] align English DBPedia abstracts with Wikidata triples and provide a corpus of 11 million high quality alignments. They use adhoc pipelines of entity linking, coreference resolution and string matching based predicate linkers. In our experiments, we find that our end-to-end linker, REFCOG outperforms such pipeline systems. In another line of work, multilingual fact retrieval [Jiang et al., 2020] is used to judge the factual knowledge captured within LM parameters by predicting masked entities in facts. However, we are only concerned with retrieving the facts present in input text.

## 3. Multilingual Fact Linking: Problem Overview

A knowledge graph (KG) contains a list of entities $E$, relations $R$ and facts $F$, where the *i*th fact links two entities ($e_x, e_y \in E$) with a relation ($r_y \in R$) and is defined as, $F_i = (e_x; r_y; e_z)$. Multilingual KGs like Wikidata also contain textual labels of entities and relations in multiple languages. Given the set $L$ of languages in the KG, for each language $l$, we construct the fact label by concatenating the labels of the entities and relation, if available. The label of $F_i$ in language $l \in L$, $F_i^l$, exists if the entity and relation labels are available in language $l$, i.e., if $e_x^l$, $r_y^l$ and $e_z^l$ exist in KG, then, $F_i^l = (e_x^l; r_y^l; e_z^l)$. [2] For example, in Table 1, (Q12146; P17; Q39) and (Landschaft; country; Switzerland) represent the fact and it's English label.

Multilingual Fact Linking (MFL) aims to discover the set of linked KG facts, $LinkedFacts(T_m) \subseteq F$ that are explicitly expressed in text $T$ of language $m$. The output set can be formally defined as ($\Rightarrow$ is used to represent the fact expressed in text):

$$\text{LinkedFacts}(T_m) = \{F_i : F_i = (e_x; r_y; e_z) \ \wedge F_i \in F \ \wedge \ T_m \Rightarrow F_i\}. \tag{1}$$

In order to predict this set, the fact labels available in different languages are used.

In Section 4, we describe the INDICLINK dataset curated for the MFL task and in Section 5, we discuss the baselines and proposed models for the task.

---

2. For right to left languages like Urdu, we use $(e_z^l; r_y^l; e_x^l)$ as the fact description.

| IndicLink | English (EN) | Hindi (HI) | Telugu (TE) | Tamil (TA) | Urdu (UR) | Gujarati (GU) | Assamese (AS) | Total |
|---|---|---|---|---|---|---|---|---|
| #Test Examples | 1002 | 889 | 888 | 881 | 1001 | 881 | 887 | 6429 |
| #KG Facts | 4.6M | 230K | 145K | 248K | 361K | 91K | 257 | 4.6M |

Table 2: The new INDICLINK dataset (Section 4) contains examples in English and corresponding manually translated test examples in six Indian languages. KG facts labels are always available in English but only sparsely available in other languages.

## 4. IndicLink: A New Dataset for Fact Linking in Indian Languages

For curating a fact linking dataset, we need a collection of sentences and the KG facts expressed in them. We use subset of Wikidata facts as the oracle KG facts by considering all facts that exist between top 1 million Wikipedia entities. Existing entity linking datasets only provide the mentioned entities, but we also need the relations between entities. So we re-purpose a relation classification dataset for fact linking by collecting the entity pair that express the particular relation.

We use WebRED [Ormándi et al., 2021] for this purpose as it is the largest relation classification dataset covering over 100 relations. Multiple relations associated with a sentence are kept as separate examples in WebRED. However, not all examples have associated Wikidata entities. In some cases, the relation maybe expressed between entity mentions that are literals such as dates/numbers or refer to entities that don't exist in Wikidata. Therefore, for each sentence we collect facts from examples that mention a relation between valid Wikidata entities in the sentence. We associate a special NULL fact for sentences which have no expressed relations.

WebRED contains sentences only in English. We extend it to multiple languages by translating the test sentences using professional translators. To ensure high quality of translations, we use 3 layers of quality checks - initial translation, review and proof-reading.

To encourage research in Indian languages, which have historically lacked knowledge-linked resources, we consider six Indian languages – Hindi, Telugu, Tamil, Urdu, Gujarati and Assamese – for our multilingual fact linking dataset, INDICLINK. Table 2 contains the number of test examples and KG facts considered. For 6,429 sentences we end up with 11,293 facts implying an average of 1.7 linked facts per sentence. We show some examples in Appendix A.

We explore automated techniques for getting training examples as it is expected that high quality language-specific training data will be unavailable for the vast majority of languages. We translate 31K WebRED training sentences into the respective Indian languages using Google Translate. The one exception is Assamese which does not have a translation system available.

## 5. REFCOG: Proposed Method for MFL

### 5.1 Dual Encoder - Cross Encoder architecture

The Dual Encoder - Cross Encoder architecture is commonly used for tasks such as semantic search [Reimers and Gurevych, 2019a], and more recently for tasks like Entity Linking [Botha et al., 2020] that require classification over a large target space. It typically involves a pipeline of dual encoder based retrieval model followed by a cross encoder re-ranking model to get the most relevant targets for the given input text. The retrieval model returns the top–$k$ targets from the entire set. Then the re-ranking model scores the top–$k$ targets using a slower model that would have been intractable

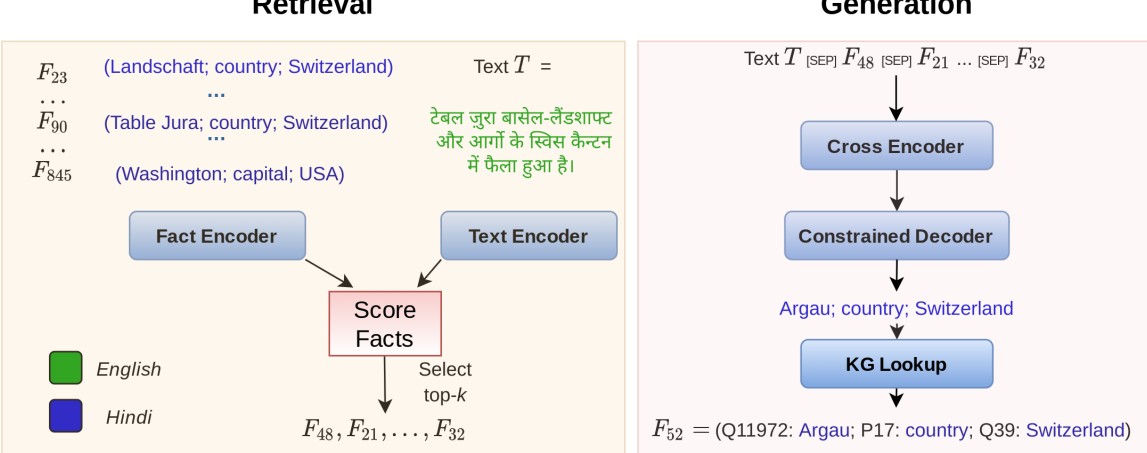

Figure 2: REFCOG architecture for linking Hindi sentences with KG facts (using their English labels). Fact-Text Dual Encoder scores the text, $T$, with all the KG facts, $F_i$, and outputs the top–$k$ facts. A Generative Seq2Seq model encodes the text $T$ concatenated with top–$k$ retrieved facts. A constrained decoder is then used to generate the correct fact.

to apply on the complete set. Apart from re-ranking cross encoders, we also experiment with a Seq2Seq cross encoder. We explain the dual encoder and cross encoders used below.

## 5.2  Fact-Text Dual Encoder

Dual Encoders are generally used for neural retrieval. They encode the input and target text independently and use the cosine similarity between the embeddings to assess their relevance. This strategy can be scaled to millions of KG facts as all the facts can be encoded beforehand, independent of the input text. The input text and facts are encoded using the text encoder and fact encoder, respectively. We build an approximate nearest neighbours index (using FAISS [Johnson et al., 2019]) of the fact embeddings which can be used to retrieve the top–$k$ facts efficiently, for a given text embedding. We initialize both the encoder parameters with LaBSE [Feng et al., 2020] weights as it is pre-trained for cross-lingual text retrieval over 109 languages. In section 7.3, we explore various choices for choosing the language of fact label, $F_i^{Ret}$, that is used for retrieval.

## 5.3  Constrained Generation Cross Encoder

Seq2Seq models can be used as cross encoders that take as input the concatenated text and retrieved facts and uses a decoder to generate the most confident fact. This allows the model to produce facts that are not returned by DE as well, overcoming the issue of error propagation in re-ranking systems whose performance is limited by the facts returned by the dual encoder. To ensure that only valid KG facts are generated, following mGENRE [Cao et al., 2021], we constrain the beam search of the decoder to follow a trie constructed from the English labels of KG facts (all facts considered have an English label available). At every step of decoding, the vocabulary is restricted to the words allowed by the trie to follow the prefix generated so far. We note that the current formulation is optimized for generating a single best fact and we leave it to future work to extend it to produce a set of facts. For now, we consider the top–$b$ facts resulting from the beam search as the final system output.

$$\text{COGEN}(T) = \text{TRIE-DEC}(\text{CE}(T||F_{T1}^{Rnk}\ldots||F_{Tk}^{Rnk})) \tag{2}$$

We initialize the Seq2Seq weights from the trained mGENRE model. For fair comparison, we use the same initialization for the cross-encoders in the classification models as well.

**KG Lookup**: We maintain a dictionary of the fact id and corresponding fact label (constructed from entity and relation labels). We take the predicted fact label and look up this dictionary to get the corresponding fact id. In case constraints are not applied, then decoder may generate invalid fact labels not present in the dictionary.

We refer to DE+CoGen model as Retrieval based Fact-Constrained Generation (REFCOG). A schematic of the model is shown in Figure 2.

### 5.4 Alternative Cross Encoders: Re-ranking

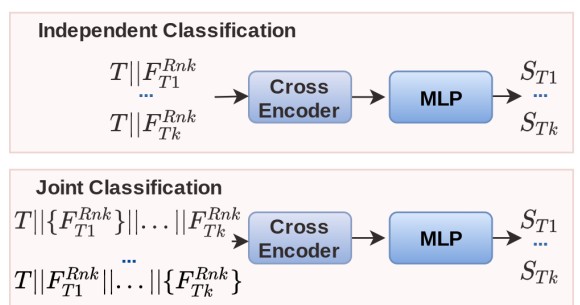

Figure 3: Independent and Joint Classification for re-ranking the facts output by the retrieval model.

Since dual encoders encode the source and target independently, they fail to capture fine-grained interactions between them. In prior work [Botha et al., 2020], classification-based cross encoders are often used to re-score the retrieved results. The re-scoring is done by concatenating input text with each of the retrieved results, allowing for inter-attention between the input text and fact label.

We explore two classification-based based cross encoder architectures (shown in Figure 3) for re-ranking the top–$k$ facts returned by a retrieval model ($F_{T1}, F_{T2}..F_{Tk}$).

**Independent Classification**: The input text $T$ is concatenated with the textual descriptions of the retrieved facts, $[F_{Ti}]^{Rnk}$ (the language chosen for ranking may be different from that used for retrieval, $[F_{Ti}]^{Ret}$) and passed through the cross encoder. An MLP operates on the pooled embedding to produce a score between 0 and 1 indicating it's confidence on whether the fact is expressed in the text or not. The scoring function can be expressed as,

$$\text{INDCLS}(F_{Ti}) = \text{MLP}(\text{CE}(T||F_{Ti}^{Rnk})) \tag{3}$$

In this case, each fact is considered independently and does not make use of the dependencies that may exist among the facts.

**Joint Classification**: In order to score the facts jointly [Kolluru et al., 2021], the score of the $i$th fact, $F_{Ti}$ is computed by concatenating the text $T$ along with all the fact labels. The $i$th fact is specially marked with a beginning { and ending }, so that the model is aware of the fact that has to be scored. This concatenated string is embedded using the cross encoder. Similar to Independent Classification case, the MLP then scores the pooled embeddings. Since the embeddings are pooled from all the facts, the model can capture the dependencies among facts as well.

$$\text{JNTCLS}(F_{Ti}) = \text{MLP}(\text{CE}(T, F_{T1}^{Rnk}||\ldots||\{F_{Ti}^{Rnk}\}||\ldots||F_{Tk}^{Rnk})_{F_{Ti}}) \tag{4}$$

## 6. Experimental Setting

**Dataset**: As described in Section 5, we use the auto-translated examples of WebRED (total 186K examples with 31K in EN, HI, TE, TA, UR, GU and 0 for AS) to train the model and the manually translated test examples of INDICLINK for evaluation. We randomly choose 5% of the training examples to be used as validation set for early stopping, during model training.

**Evaluation Metrics**: We compare the facts predicted by the model with the gold set of facts and report the value of Precision@1 (P@1) and Recall@5 (R@5). P@1 is the fraction of examples where the most confident fact is contained in the gold set.[3] R@5 is the fraction of gold facts that are present in the top-5 predicted facts. We also compute **macroP@1**, the macro-average counterpart to P@1, where the gold facts are divided into relation-specific classes, performance computed independently in each class and then averaged across all classes.

**Implementation**: We implement all models in Pytorch framework. We use Sentence Transformers [Reimers and Gurevych, 2019b] library for training dual encoder models and use GENRE codebase[4] and fairseq [Ott et al., 2019] library for implementing the various cross encoders. We train the dual encoder and generation models for 5 epochs each. Total training time is 6 hrs on A100 GPU.

## 7. Experiments

We conduct experiments to address the following three questions:

- How well does REFCOG, a retrieve+generation architecture, work for Multilingual Fact Linking, especially when compared to retrieve+reranking models for the task? (Section 7.1)

- What is the effect of different components in generative decoding? (Section 7.2)

- How to effectively utilize multilingual fact labels during retrieval as well as generative decoding stages of REFCOG? (Section 7.3)

We note that translation systems may not be available at inference time for certain languages. For example, Assamese, one of the languages we consider in our experiments, is currently not supported by Google Translate. Hence, in our experiments, we aim to evaluate multilingual fact linking by relying on the cross-lingual ability of the model, rather than test-time translation.

### 7.1 Effectiveness of REFCOG

In Table 3, we compare the results of various models trained on INDICLINK. All the cross encoder models use English fact labels and $DE_{ALL-Sum}$ for retrieval (which is explained in detail in Section 7.3). We notice that the DE models particularly struggle with retrieving NULL fact, as its label ("None") does not have any word overlap with the sentence. Therefore, in INDCLS-N and JNTCLS-N, we deterministically add NULL to the input, along with the remaining DE outputs. This is not required for the REFCOG model as it is free to generate facts even if they are not returned by the retrieval module. REFCOG model outperforms classification based re-ranking by 10.7, 15.2 pts in P@1, R@5, respectively. Within the re-ranking models, INDCLS achieves better performance compared to JNTCLS, indicating that providing the other facts tend to confuse the model.

---

3. NULL is also considered as a separate fact for measuring performance.
4. github:facebookresearch/GENRE

| Model | EN | HI | TE | TA | UR | GU | AS | Average | |
|---|---|---|---|---|---|---|---|---|---|
| | P@1 | P@1 | P@1 | P@1 | P@1 | P@1 | P@1 | P@1 | R@5 |
| DE$_{\text{ALL-Sum}}$ | 37.5 | 29.7 | 32.8 | 27.8 | 28.7 | 29.9 | 13.8 | 28.6 | 31.9 |
| +INDCLS | 26.3 | 20.9 | 23.5 | 20.3 | 20.1 | 22.7 | 9.6 | 20.5 | 31.9 |
| +INDCLS-N | 46.2 | 41.8 | 44.3 | 42.5 | 43.8 | 40.9 | 29.5 | 41.4 | 36.3 |
| +JNTCLS | 13.8 | 13.8 | 12.7 | 13.1 | 11.9 | 14.9 | 5.9 | 12.3 | 31.9 |
| +JNTCLS-N | 38.5 | 38.7 | 39.9 | 38.4 | 38.4 | 38.6 | 34.0 | 38.1 | 36.3 |
| REFCOG$_{\text{ALL-Sum, EL}}$ | **56.4** | **52.4** | **53.4** | **53.2** | **53.6** | **52.5** | **43.1** | **52.1** | **51.5** |
| -Constraints | 55.9 | 52.3 | **53.4** | 52.8 | 53.4 | 52.4 | 42.3 | 51.9 | 49.5 |
| -SRO links | 42.0 | 38.8 | 38.9 | 35.8 | 38.2 | 36.9 | 32.4 | 37.6 | 21.4 |
| -DE | 50.2 | 48.7 | 49.1 | 47.4 | 47.5 | 47.7 | 39.6 | 47.2 | 45.6 |

Table 3: Comparison of different models on the INDICLINK dataset. REFCOG with ALL-Sum dual encoder labels and EL cross encoder labels, outperforms independent (INDCLS) and joint (JNTCLS) classification based re-ranking on top of DE$_{\text{ALL-Sum}}$. Ablations indicate the importance of DE and joint prediction of S, R and O for the REFCOG model. Constraints reduce the P@1, R@5 metrics but ensure production of only valid facts. Please see Section 7.1 and Section 7.2 for further details.

## 7.2 REFCOG ablations

In Table 3, we consider four variants of the REFCOG model: (1) REFCOG w/o Constraints: after removing the constraints on the decoder beam search, (2) REFCOG w/o SRO links: predicting the subject (S), relation (R) and object (O) of the fact independent of one another, (3) REFCOG w/o DE: removing DE facts from Cross Encoder input, and (4) REFCOG w/o DE, Constraints: removing DE facts and constraints.

Removing constraints leads to generation of incorrect facts in 865/6253 examples and reduces performance by 0.2 pts in P@1 and 2 pts R@5, respectively. Instead of predicting the entire fact jointly, we perform an ablation in which we predict each of the components independently. This leads to reduction in performance of 18.1 pts in P@1, showing the importance of jointly predicting all the components. Unlike the re-ranking models, REFCOG can generate facts even in the absence of Dual Encoder retrieved facts. This allows us to evaluate the model performance without the Dual Encoder. The results indicate that DE facts are responsible for (4.9, 5.9) pts improvement in P@1, R@5.

## 7.3 Effect of Multilingual Fact Labels

We consider fact labels in English (EL), the language of input text $T$ - Text Language (TL), English and TL combined (ETL), and all the languages in IndicLink (ALL).

For the dual encoder, we either form a single embedding for the fact by concatenating the labels in various languages or embed the different language labels separately. In either case, we only consider languages for which the fact has a label available. The score for a fact $D$ is computed as the cosine similarity between the text embedding and fact embedding. In case multiple embeddings are associated with a fact, we aggregate their individual scores either through a sum/max operation. The various types of fact labels and scoring operations can be summarized as follows:

- EL (or TL): Using the English (or Text language) label of the fact.

| Model | Fact Label Selection | EN | HI | TE | TA | UR | GU | AS | Average | |
|---|---|---|---|---|---|---|---|---|---|---|
| | | P@1 | P@1 | P@1 | P@1 | P@1 | P@1 | P@1 | P@1 | R@5 |
| DE | EL | 27.2 | 19.8 | 20.8 | 16 | 19.1 | 19.6 | 8.3 | 18.9 | 26 |
| | TL | 24.3 | 13.3 | 14.0 | 10.2 | 12.2 | 12.1 | 5.1 | 13.2 | 20 |
| | ETL-Concat | 26.6 | 21.3 | 25.6 | 17.5 | 23.6 | 22.7 | 6.9 | 20.8 | 28.4 |
| | ALL-Concat | 28.2 | 19.9 | 21.1 | 19.1 | 21.2 | 19 | 7.5 | 19.6 | 27 |
| | ETL-Max | 27.8 | 24.1 | 24.9 | 20.2 | 21.9 | 23.0 | 5.4 | 21.2 | 28.9 |
| | ETL-Sum | 27.8 | 25 | 26.4 | 20.7 | 21.2 | 23.5 | 6.2 | 21.6 | 29.3 |
| | ALL-Max | 31.1 | 25.8 | 25.5 | 22.2 | 24.1 | 24.9 | 10.1 | 23.4 | 26.1 |
| | ALL-Sum | 37.5 | 29.7 | 32.8 | 27.8 | 28.7 | 29.9 | 13.8 | 28.6 | 31.9 |
| REFCOG | EL | 56.4 | **52.4** | 53.3 | **53.2** | **53.6** | 51.9 | **43.1** | **52.1** | **51.5** |
| | TL | 55.2 | 47.2 | 48.8 | 47 | 47.6 | 47.8 | 33.3 | 46.9 | 43 |
| | ETL-Concat | 57 | 49.7 | **53.5** | 49.9 | 51.8 | 50.2 | 41 | 50.6 | 50 |
| | ALL-Concat | **57.1** | 51.5 | 53.3 | 50.4 | 51.1 | **52.1** | 40.1 | 50.9 | 50 |

Table 4: Multilingual fact labels in Retrieval and Generation models (Section 7.3). EL, TL, ETL and ALL correspond to descriptions in English, language of input text $T$, EL+TL and all languages, respectively. Concat, Max and Sum refer to concatenation, max and sum scoring operations. For REFCOG, we use ALL-Sum fact labels for retrieval and experiment with cross encoder fact labels.

- ETL-Concat (or ALL-Concat): Using the concatenated English and Text language label of the fact (or concatenation of all language labels available).

- ETL-Max (or ALL-Max): Embed the labels in each language in ETL (or ALL) separately and consider the max of their cosine similarity as the score for the fact.

- ETL-Sum (or ALL-Sum): Use the sum of scores of labels in each language in ETL-Sum (or ALL-Sum).

We find that embedding the fact labels separately and adding their individual scores leads to the best performance across all languages in P@1, R@5.

| Fact Label | Complete | | Subset | |
|---|---|---|---|---|
| | P@1 | macroP@1 | P@1 | macroP@1 |
| EL | **52.1** | 12 | 65.2 | 35.4 |
| TL | 43 | 6.1 | 63.7 | 43 |
| ETL-Concat | 50.6 | **15.7** | **67.5** | **55** |
| ALL-Concat | 50.9 | 14.7 | 64.2 | 54.8 |

Table 5: P@1, macroP@1 of REFCOG with fact labels in various languages at cross encoder stage. The macroP@1 is evaluated for the Complete test set as well as the Subset where descriptions are available in all languages. Improvement in macroP@1, indicates stronger performance on facts with less-frequently occurring relations.

However, for cross encoders, we don't find similar improvements on concatenating the various language labels. To study this further, we also compute the macroP@1 in Table 5. The results on the Complete test set indicates that using language fact labels other than English does not help in P@1 but result in a modest increase of 3.7 pts in macroP@1 for ETL-Concat. This indicates that other language fact labels can help in facts that involve less-frequently occurring relations. Also, language fact labels are not consistently available in all languages. So we construct a modified test set that uses only KG facts where fact labels are available in all languages and the test examples that can be answered with these KG facts. On this subset,

ETL-Concat shows an increase of 19.6 pts in macroP@1, compared to using only English fact labels. This shows that robustly handling sparse availability of fact labels can improve performance. ALL-Concat does not seem to improve performance over ETL-Concat which may be due to larger sequence lengths when all language labels are concatenated.

### 7.4 REFCOG Error Analysis

We analyze the errors made by REFCOG and classify it into the following three types:

- **Rare relations**: The model struggles with facts that contain rare relations which can be traced back to WebRED data where the top-10 relations are responsible for 80% of the examples.
- **NULL fact**: The model wrongly predicts NULL in 33% of the examples, demonstrating the difficulty in detecting absence of any facts.
- **Issues with Gold**: Few examples contain relations that are not explicitly implied by the sentence but require background world knowledge.

We show sample model predictions in Appendix A.

## 8. Conclusion

In this work, we introduce the task of multilingual fact linking and present a new evaluation dataset INDICLINK containing examples in English and six Indian languages. We explore various dual encoder, cross encoder architectures and find that the proposed Retrieval+Generation model, REFCOG, outperforms classification-based reranking systems by 10.2 pts in P@1. We also release the marked entity mentions for each linked fact, which also makes it usable for evaluating entity linking and relation classification, forming the first such resource for Indian languages. We leave it to future work to further explore these directions.

### Acknowledgements

We thank Vidhi Kapoor, Sumit Chabbra and the Google Localization Team for their timely help with manual translations. We also thank Jan Botha for insightful discussions on the MEL system and Tom Kwiatkowski for helpful comments on an earlier version of the paper.

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

## Appendix A. REFCOG Qualitative Examples

For three examples from INDICLINK, we list the sentence in English, the corresponding translations in Indian languages and the gold facts (along with their English labels) annotated for the example. For the sentence in each language, we also list the facts output by DE and the REFCOG beam search in the order of decreasing confidence.

**Example 1**:
*Language*: English

*Sentence*: However, with both Hedda and Ingrid competing at the same Olympic Games, but in different sports, the sisters were historic in a Norwegian context.

*Gold Facts*:
(Q262950: Hedda Berntsen; P27: country of citizenship; Q20: Norway)
(Q262950: Ingrid Berntsen; P27: country of citizenship; Q20: Norway)

*DE Facts*:
(Anita Hegerland; country of citizenship; Norway)
(Astrid S; country of citizenship; Norway)
(Helge Ingstad; country of citizenship; Norway)
(Ane Stangeland Horpestad; member of sports team; Norway women's national football team)
(Icemaiden; country of citizenship; Norway)

REFCOG *Beam Search*:
NULL
(Q17107619: Astrid S; P27: country of citizenship; Q20: Norway)

हालाँकि, हेड्डा और इनग्रिड दोनों एक ही ओलंपिक गेम में मुकाबला कर रही हैं, लेकिन उनका खेल अलग-अलग है, इन दोनों बहनों ने नॉर्वे में ऐतिहासिक उपलब्धियां हासिल की हैं

Figure 4: Example 1, Hindi Sentence

(Q20: Norway; P17: Country; Q20: Norway)
(Q236632: Anna Carin Zidek; P1344: participant of; Q9672: 2006 Winter Olympics)
(Q43247: Ingrid Bergman; P27: country of citizenship; Q34: Norway)

*Language*: Hindi

*Sentence*: Please see Figure 4.

*Gold Facts*:
(Q262950: Hedda Berntsen; P27: country of citizenship; Q20: Norway)
(Q262950: Ingrid Berntsen; P27: country of citizenship; Q20: Norway)

*DE Facts*:
(Anita Hegerland; country of citizenship; Norway)
(Ann Kristin Aarønes; member of sports team; Norway women's national football team)
(Andrine Hegerberg; member of sports team; Norway women's national football team)
(2012 European Women's Handball Championship; participating team)
(Norway women's national handball team; sport; handball)

REFCOG *Beam Search*:
(Q270197: Ann Kristin Aarønes; P27: country of citizenship; Q20: Norway)
NULL
(Q264681: Camilla Herrem; P27: country of citizenship; Q20: Norway) (Q236632: Anna Carin Zidek; P1344: participant of; Q9672: 2006 Winter Olympics) (Q270197: Ann Kristin Aarønes; P1344: participant of; Q8531: 1996 Summer Olympics)

**Example 2**:
*Language*: English

*Sentence*: Microsoft Confirms MKV File Support in Windows 10.

*Gold Facts*:
(Q18168774: Windows 10; P3931: developer; Q2283: Microsoft)

*DE Facts*:
(Microsoft Windows; has edition; Windows 10)
(Windows 10; copyright holder; Microsoft)
(Final Fantasy VIII; platform; Microsoft Windows)

``Microsoftএ Windows 10"ত MKV ফাইলৰ সমৰ্থন নিশ্চিত কৰিছে।

Figure 5: Example 2, Bangla Sentence

(Windows 10; developer; Microsoft)

REFCOG *Beam Search*:
(Q18168774: Windows 10; P3931: copyright holder; Q2283: Microsoft)
NULL
(Q18168774: Windows 10; P178: developer; Q2283: Microsoft)
(Q1406: Microsoft Windows; P178: developer; Q2283: Microsoft)
(Q245006: Final Fantasy VIII ; P400: platform ; Q1406: Microsoft Windows)

*Language*: Bangla

*Sentence*: Please see Figure 5.

*Gold Facts*:
(Q18168774: Windows 10; P3931: developer; Q2283: Microsoft)

*DE Facts*:
(Microsoft Windows; has edition; Windows 10)
(Windows 10; developer; Microsoft)
(Windows 10; copyright holder; Microsoft)

REFCOG *Beam Search*:
(Q18168774: Windows 10; P3931: copyright holder; Q2283: Microsoft)
(Q18168774: Windows 10; P178: developer; Q2283: Microsoft)
NULL
(Q1406: Microsoft Windows; P178: developer; Q2283: Microsoft)
(Q2283: Microsoft; P1830: owner of; Q192527: Windows Media Player)

**Example 3**:
*Language*: English

*Sentence*: Wind Cave National Park, which is another area administered by the National Park Service, borders portions of the forest in the southeast.

*Gold Facts*:
(Q1334313: Wind Cave National Park; P137: operator; Q308439: National Park Service)

*DE Facts*:
(Wind Cave National Park; operator; National Park Service)

بوا غار قومی پارک ، جو قومی پارک سروس، کے زیر انتظام ایک اور علاقہ ہےجو جنوب مشرق میں جنگل کی کچھ سرحدوں سے متصل ہے۔

Figure 6: Example 3, Urdu Sentence

(Virgin Islands National Park; operator; National Park Service)
(Valley Forge National Historical Park; operator; National Park Service)
(Weir Farm National Historic Site; operator; National Park Service)
(Hovenweep National Monument; operator; National Park Service)

REFCOG *Beam Search*:
(Q1334313: Wind Cave National Park; P137: operator; Q308439: National Park Service)
NULL
(Q308439: National Park Service; P355: subsidiary; Q3719: National Register of Historic Places)
(Q308439: National Park Service; P749: parent organization; Q608427: United States Department of the Interior)
(Q1334313: Wind Cave National Park; P131: located in the administrative territorial entity; Q490716: Custer County)
*Language*: Urdu

*Sentence*: Please see Figure 6.

*Gold Facts*:
(Q1334313: Wind Cave National Park; P137: operator; Q308439: National Park Service)

*DE Facts*:
(Joshua Tree National Park; operator; National Park Service)
(Cuyahoga Valley National Park; operator; National Park Service)
(Kicking Horse Pass; located in protected area; Yoho National Park)
(Jewel Cave National Monument; operator; National Park Service)
(Gateway National Recreation Area; operator; National Park Service)

REFCOG *Beam Search*:
NULL
(Q152820: Cuyahoga Valley National Park; P137: operator; Q308439: National Park Service)
(Q735202: Joshua Tree National Park; P137: operator; Q308439: National Park Service)
(Q36600: The Hague; P206: located in or next to body of water;Q1693: North Sea)
(Q36600: The Hague; P131: located in the administrative territorial entity; Q694: South Holland)

