# OpenReview forum: "Multilingual Fact Linking"
_AKBC.ws/2021/Conference — AKBC 2021_

### Official Review · Reviewer_pfgr · 2021-07-19
**Solid but limited multilingual fact linking resource**

**Rating:** 6
**Confidence:** 4

**Review:**

=== Summary ===

This paper considers the task of multilingual fact linking, where the goal is to link abstract representations of facts to their language-specific representations in multiple languages. The challenge is that although "worldly facts" themselves are language-agnostic, annotations of how they appear in text form (e.g., described in a sentence or named) are often restricted to only a few languages. The proposed task of multilingual fact linking seeks to link KG facts with their mentions in multilingual text, even when the label of the fact doesn't match the language of the text (e.g., matching <NYC, country, USA> to a Hindi text that expresses this).

=== Strengths ===

- The paper is mostly well-written. It's a bit hard to follow exactly what the task setting is (which components are in language A and which are in language B). It would be much clearer if a simple, formal definition was given in the introduction. It also would be good to perhaps highlight the MFL row in table 1 as "this paper".

- Overall, the task is well motivated and the introduced dataset, despite some limitations (see below), may be helpful for further multilingual development.

- The paper proposes a retrieval + generation model (ReFCOG) that performs well as compared to retrieval + reranking models, which provides a good starting point for the task.

=== Weaknesses ===

- One of the biggest downsides is that the evaluation dataset (IndicLink) is (a) limited in typological scope to a select family of languages, and (b) translated from English. The question of "why not translate" is addressed nicely in Section 3.2 of Clark et. al. 2020 (https://arxiv.org/pdf/2003.05002.pdf). Thus, there is a concern that the artifacts introduced by "translationese" may considerably hurt generalization performance to purely native text.

- The proposed ReFCOG model is composed of several existing components previously proposed in multi-lingual work, and seems more like a baseline than a purely "novel" approach. That's ok, as it's good to have documented performance of simple to slightly complex models when introducing a new dataset (but the messaging might be better off slightly altered to reflect that).

=== Questions ===

- Why does constrained generation *reduce* performance? It seems that by restricting generation to only valid facts, it wouldn't be filtering any beam candidates that reflect possible correct answers. So, for any unconstrained generation that would be correct, why would it not still be the top candidate from the constrained search? It would seem that the set of correct answers should be a perfect subset.

=== Missing references ===

- The related work could be more complete in included related work from multilingual fact retrieval, for example, Jiang et. al. (https://aclanthology.org/2020.emnlp-main.479.pdf).

---

> ### Author Response · Authors · 2021-07-29
> **Choice of languages, Novelty and Constrained generation issue**
>
> **Effect of Translationese artifacts**:
>
> We deliberated on the choice of native text vs translationese during IndicLink design, and decided to go ahead with translations noting the effort spent in creating WebRED (Ormandi et. al., 21) for English sentences and the difficulty in finding experts who can do the same in other languages. Therefore, we decided to proceed with translating already available WebRED English sentences into Indian languages. This allowed us to extend to more languages than would be otherwise possible.
>
>
> **Lack of non-Indian languages**:
>
> Indian languages have been historically underserved in the NLP community, particularly for Information Extraction tasks. In order to make progress on this front, we decided to use our available budget to exclusively focus on Indian languages. Apart from fact linking, IndicLink forms the first resource for evaluating important IE tasks such as entity linking and relation classification for web text in Indian languages.
>
> **Novelty**:
>
> The ReFCoG model demonstrates the benefits of augmenting generation with retrieval models for IE tasks. Prior entity linking systems (Botha et. al., 20) relied on retrieval based models while recent works (Nicola et. al., 21) have proposed constrained generation models as a replacement to retrieval models.
>
> ReFCoG is the first system to successfully demonstrate that both the approaches can be combined to complement each other and achieve strong performance. We believe that this is a general observation that can also benefit entity linking. This can serve as a starting point for exploring end-to-end ways of combining both the retrieval and generation paradigms in the context of entity linking.
>
> **Issue with constrained generation**:
>
> We agree that the correct facts output by unconstrained generation must be a proper subset of the constrained generation and the current results are contradictory to this. On a closer look, we uncovered that the trie used for the constrained system was incomplete (missing some valid fact labels) and so the unconstrained system was allowed to produce more valid facts than the constrained system. Once we correct this issue, constrained generation does indeed outperform the unconstrained model by 0.6 P@1 and 2 R@5. We have updated these results in Table 3 and Table 4 of the latest draft. We note that all the other presented conclusions remain the same.
>
> **Presentation and References**:
>
> We thank you for pointing out the challenges involved in understanding the task formulation in the submitted version. We have added a better description of the task in the introduction, clarified the MFL task in Table 1 and simplified the formulation in Section 3 in the latest uploaded draft.
>
> We have added the Jiang et. al., 2020 reference but we note that the work addresses a fundamentally different problem of “retrieving” facts from the memory of pretrained LM. Instead, we focus on retrieving only a subset of facts that are relevant to the sentence.
>
> Botha, J. A., Shan, Z., & Gillick, D. (2020). Entity linking in 100 languages, EMNLP, 2020
> De Cao, N., Izacard, G., Riedel, S., & Petroni, F. (2020). Autoregressive entity retrieval, ICLR, 2021
> Ormandi, R., Saleh, M., Winter, E., & Rao, V. (2021). WebRED: Effective Pretraining And Finetuning For Relation Extraction On The Web. arXiv preprint arXiv:2102.09681.

---

### Official Review · Reviewer_dGrx · 2021-07-21
**Good multilingual fact linking corpus with interesting model, but missing an obvious baseline**

**Rating:** 6
**Confidence:** 4

**Review:**

This paper studies the problem of fact linking -- linking facts from a knowledge graph (KG) to a given sentence where the fact is mentioned. More specifically, recognizing the sparsity and skewness of KG entity and relation labels in terms of language coverage, this paper proposes to study the multilingual setting, where the input sentence is in an underrepresented language and one needs to link KG facts, often only expressed in more high-resource languages, to the input sentence. A new test set for 6 Indian languages is created by manually translating sentences from WebRED with professional translators. The paper also proposes a model for this task based on the Dual Encoder - Cross Encoder architecture, but unlike conventional Cross Encoder, it follows mGENRE and proposes a constrained autoregressive decoder to generate valid facts. It is shown that the proposed model outperforms re-ranking based Cross Encoders by a large margin on the new test set.

Strengths
- A new problem and dataset with an emphasis on underrepresented languages in NLP study
- Strong baseline model with autoregressive decoder for fact linking
- Empirical results seem promising

Weaknesses
- An obvious baseline is missing, where one uses some off-the-shelf machine translation tool to translate the English labels in the KG to the target language, which effectively turns the multilingual problem into monolingual. Discussion and demonstration of the inefficacy (if there's any) of that is needed.
- It's surprising that constrained generation leads to worse performance than free-form generation, and there lacks a convincing analysis and explanation for that. In principle, the constrained version should be strictly better than the free-form version, unless there are unnecessary constraints or something happens with the optimization. Since it's a major contribution of this paper, it merits more thorough discussion.

Minor
- Table 2, missing "K" in the Assamese row?
- Related work, "Fact Extraction", "whereas fact extraction system do not":  "system" -> "systems"
- I find it a bit odd to capitalize phrases like "Entity Linking", "Coreference Resolution", and "Multilingual Fact Linking" in the main text. Probably better to be lower-cased.

---

> ### Author Response · Authors · 2021-07-29
> **Machine Translation baseline, Constrained generation issue**
>
> **Machine Translation baseline**:
>
> We thank the reviewer for pointing out this important baseline. However, we note that this requires access to language-specific resources such as a translation system which may not be readily available at inference time or may be expensive to use. Translating the 4.5 million English facts into 6 languages can be rather challenging. Moreover, some low-resourced languages may not even have translation systems readily available. For example, among the IndicLink languages, Google Translate does not support Assamese. Therefore, in line with other multilingual tasks like TyDiQA (Clark et. al., 20), we hope to approach this problem by relying on the cross-lingual ability of the model.
>
> **Issue with constrained generation**:
>
> We agree that the constrained version is expected to be strictly better than the free-form version and the current results are contradictory to this. On a closer look, we uncovered that the trie used for the constrained system was incomplete (missing some valid fact labels) and so the unconstrained system was allowed to produce more valid facts than the constrained system. Once we correct this issue, constrained generation does indeed outperform the unconstrained model by 0.6 P@1 and 2 R@5. We have updated these results in Table 3 and Table 4 of the latest draft. We note that all the other presented conclusions remain the same.
>
> **Minor Suggestions**:
>
> We have incorporated the suggested changes in related work and removed capitalization of tasks like entity linking, relation classification in the latest draft. We note that in Table 2, there are indeed only 257 Assamese labels for facts, stemming from the lack of Wikidata fact labels in low-resourced languages.
>
>
>
> Clark, J. H., Choi, E., Collins, M., Garrette, D., Kwiatkowski, T., Nikolaev, V., & Palomaki, J. (2020). TyDi QA: A benchmark for information-seeking question answering in typologically diverse languages. Transactions of the Association for Computational Linguistics, 8, 454-470.

---

### Official Review · Reviewer_F5Kc · 2021-07-21
**Interesting task and systems for multilingual fact linking, some shortcoming**

**Rating:** 6
**Confidence:** 3

**Review:**

The contribution of this paper is two-fold.
First, the authors create (and make available) IndicLink, a dataset of multilingual fact linking in 5 indian languages + English. The dataset is translated from English, from a portion of WebRED by annotators. The authors attempt to coin "multilingual fact linking" and propose a formalization of the problem.
Second, the authors evaluate current bi-encoder and cross-encoder retrieval and reranking architectures for fact linking using an multilingual transformer model to encode facts (subject, predicate and object names) and additionally
propose a system following the mGenre paradigm where in the standard bi/dual encoder + reranking cross-encoder, the reranking cross-encoder is replaced by a generative (constrained) decoder.
The authors propose an evaluation of their system on their dataset, against other current systems on all 5 languages and english and also perform various ablations of the model in order to determine the effect of the different components on the final results.


Strong points:
- Interesting system that shows promising results.
- A new dataset that can be of interest to the community.
- Relatively well written

Shortcomings:
 - Potential bias in the dataset construction due to the translation from English as pointed out by another reviewer.
 - Poor justification for the choice of the specific Indian languages used in the dataset. Some of the languages belong to different sub-families, which is good, but the selection criteria aren't clear. For example if the criterion was the number of speakers, then surely, a language like Bangla would have to be included. Other European languages could be included too as they are available in the source dataset.
 - The formalization for the task is difficult to parse and perhaps too arcane (doesn't follow usual conventions). As such the textual description is much easier to understand.
 - No error analysis (an appendix shouldn't be used to finish the paper, the error analysis has as much a place in the paper itself as the ablation studies.

Detailed comments and questions:
- Please clearly state the definition of "fact" for the task of fact interlinking. "Fact" has many conflicting and very different definitions in related subfields of computer science, even though the venue is clearly oriented towards fact as a synonym for assertion in a knowledge base, it's still important to say so in the introduction.
- Please improve the formalization, that's a very strange way of modeling binary relations (predicates are a binary relation between subject and object). The set of facts is clearly contained in the cartesian product of R \subseteq ExE, why isn't it defined as such? Equation 1 doesn't need to be so complex, if a fact is synonymous with triple, you can just state {F_i \in R| T_m \entails F_i}.
Since there is no formal way of defining that a "fact" is contained or expressed in a textual fragment or an utterance, you could very well use the entailment symbol (just a suggestion, this is far from being the only way). It's also likely that a diagram would go a long way towards explaining the task in a precise yet clear manner.
- What prevents the application of the methods to standard monolingual datasets for fact linking? An external benchmark beside the authors' own dataset would strengthen the approach.
- Tables 3 onwards aren't placed optimally and are far from where they're commented.
- Equations 3 and 4 would again benefit from being represented schematically.

---

> ### Author Response · Authors · 2021-07-29
> **Choice of languages, Monolingual fact linking and improved presentation**
>
> **Effect of Translationese artifacts**:
>
> We deliberated on the choice of native text vs translationese during IndicLink design, and decided to go ahead with translations noting the effort spent in creating WebRED (Ormandi et. al., 21) for English sentences and the difficulty in finding experts who can do the same in other languages. Therefore, we decided to proceed with translating already available WebRED English sentences into Indian languages. This allowed us to extend to more languages that would be otherwise possible.
>
> **Lack of non-Indian languages**:
>
> Indian languages have been historically underserved in the NLP community, particularly for Information Extraction tasks. In an effort to remedy this situation, we decided to use our available budget to exclusively focus on Indian languages. Apart from fact linking, IndicLink forms the first resource for evaluating important IE tasks such as entity linking and relation classification for web text in Indian languages.
>
> **Choice of Indian languages**:
>
> We chose the six Indian languages based on diversity of language families and geographical distribution:
> 1. Dravidian languages: Telugu, Tamil in South India,
> 2. Indo-Aryan languages: Hindi, Urdu from North/Central India, Assamese and Gujarati from East and West, respectively.
> https://en.wikipedia.org/wiki/Indo-Aryan_languages#Groups provide a detailed geographical classification of Indo-Aryan languages.
>
> We particularly choose low-resourced Assamese (same sub-family as Bangla) because of the lack of available machine translation support in Google Translate, allowing us to evaluate the zero-short performance of the ReFCoG model on Assamese sentences.
>
> **Application to Monolingual Fact Linking**:
>
> We do apply the ReFCoG model to fact linking where both the input sentence and the facts labels are of the same language (Table 4: EL fact labels and  EN text). However, we note the difficulty in comparing with prior datasets that are designed primarily for fact extraction and not linking.
>
> For example, SemEval 2010 is used to extract facts from the sentence. But to pose it as fact linking, we need an oracle set of facts from which we can choose the correct subset relevant for the sentence. These facts don’t belong to a KG like Wikidata or DBPedia and hence are difficult to compare.
> On the other hand, T-Rex facts do belong to Wikidata but are annotated using distant supervision and hence don’t necessarily represent the ground truth. WebRED overcomes all of these issues and hence we apply our technique on WebRED examples.
>
> **Presentation**:
>
> We would like to thank the reviewer for pointing out ways to improve the presentation in the paper. We have incorporated the suggested changes in the revised draft of 1. improving the task formalization, 2. adding error analysis, 3. clarifying fact definition, 4. schematic representation of alternative cross encoder, 4. adjustment of tables.
>
> Ormandi, R., Saleh, M., Winter, E., & Rao, V. (2021). WebRED: Effective Pretraining And Finetuning For Relation Extraction On The Web. arXiv preprint arXiv:2102.09681.

---

### Decision · Program_Chairs · 2021-08-18

**Decision:**

Accept

**Comment:**

This paper proposed a valuable resource in underrepresented languages, hence an exciting contribution for AKBC community. The new dataset presents a multi-lingual fact linking task for 5 indian languages in addition to English. The paper proposes a retrieval + generation model (ReFCOG) that performs well as compared to retrieval + reranking models, which provides a good starting point for the task. Reviewers have presented excellent list of suggestions to improve the paper further, especially, adding simple MT baseline (or a discussion if not possible in this setup), including systematic error analysis in the main paper, clarifying novelty of the proposed approach. I urge authors to address these in the updated draft of this paper.